# Melioidosis Knowledge Awareness in Three Distinct Groups in the Tropical Northern Territory of Australia

**DOI:** 10.3390/tropicalmed9040071

**Published:** 2024-03-28

**Authors:** Madusha P. Weeratunga, Mark Mayo, Mirjam Kaestli, Bart J. Currie

**Affiliations:** 1Menzies School of Health Research, Charles Darwin University, Darwin, NT 0811, Australia; madusha.weeratunga@nt.gov.au (M.P.W.); mark.mayo@menzies.edu.au (M.M.); mirjam.kaestli@menzies.edu.au (M.K.); 2Infectious Diseases Department, Royal Darwin Hospital, Darwin, NT 0810, Australia; 3Northern Territory Medical Program, Flinders and Charles Darwin University, Darwin, NT 0810, Australia

**Keywords:** melioidosis, melioidosis knowledge, melioidosis prevention, cultural knowledge, knowledge disparities, health literacy, health literacy

## Abstract

Melioidosis is a potentially life-threatening infection. This study aimed to assess the melioidosis knowledge among distinct participant groups in the tropical Top End of the Northern Territory (NT) of Australia. Participants were categorised into three groups: NT medical students and health research staff (Group 1: Hi-Ed), Aboriginal Rangers and Aboriginal Healthcare Workers (Group 2: Rangers/AHWs), and patients with a history of melioidosis infection (Group 3: Patients). A questionnaire was developed to collect data on demographics, risk and protective factor awareness, and knowledge acquisition sources. We used responses to calculate indices for risk knowledge (RKI), protective knowledge (PKI), overall melioidosis knowledge (MKI), and information sources (ISI). We found that 93.6% of participants in Group 1 (Hi-Ed) said that they had heard of melioidosis, followed by 81.5% in Group 3 (Patients), and 72.0% in Group 2 (Rangers/AHWs). Group 1 (Hi-Ed) participants demonstrated greater knowledge of risk-increasing behaviours but had gaps in knowledge of clinical risks like diabetes. Multiple regression revealed that the number of resources used was the only significant predictor of MKI. There are varying melioidosis knowledge levels across different NT participant groups. Targeted educational interventions are needed to enhance melioidosis awareness. A weblink with an interactive summary of our analysis can be found under Results part.

## 1. Introduction

Melioidosis, caused by the bacterium *Burkholderia pseudomallei* found in soil and water, has long been a significant health concern in Australia’s Northern Territory (NT) [1]. It is primarily transmitted to humans through percutaneous inoculation, often via open wounds. Inhalation of aerosolised bacteria can occur during severe weather events, and ingestion of untreated water containing the organism is also recognised [2,3]. The main risk factors for melioidosis infection include diabetes mellitus, hazardous alcohol use, chronic lung disease, and chronic renal disease [4].

Despite recent findings linking *B. pseudomallei* to an ancient common ancestor from the Australian environment [5], there is no documented evidence suggesting melioidosis infection among Aboriginal people before the arrival of the white settlers. For millennia, the Australian First Nations maintained a traditional hunter-gatherer lifestyle with a strong verbal history, including songlines about illnesses and avoidance strategies for “sickness country” lands [6]. No such narratives exist for melioidosis despite the pathogen predating colonization. Unlike diseases introduced through colonization, melioidosis was unmasked by it. The first reported Australian case in 1950 [7] marked the onset of the still-increasing incidence of melioidosis in northern Australia, which has disproportionately affected First Nation Australians [8]. This rise is linked to introduced sugar, alcohol, and cigarettes, with the associated chronic disease risk factors known to underlie most cases of melioidosis. Despite the high incidence, literacy around melioidosis and its prevention is limited in the general community of northern Australia, including the absence of a word for melioidosis in any Aboriginal language.

Australian First Nation people are often disadvantaged owing to complex socioeconomic factors, including reduced access to both education and healthcare services [9]. Remote living includes frequent contact with tropical soils and surface water during daily activities, such as hunting and food gathering. These factors, along with increased health co-morbidities [2], make them more susceptible to infection with *B. pseudomallei* and more likely to progress to clinical disease (i.e., melioidosis). Studies suggest Aboriginal and Torres Strait Islander Australians in the Northern Territory face a threefold greater risk of melioidosis compared to the larger Australian community [10].

A survey of patients at risk of melioidosis in Far North Queensland revealed that only 19% had heard of it [11]. A separate study from the same region documented that over 50% of melioidosis cases comprised Aboriginal Australians, even though they comprised just 14.6% of the local population [12]. This study also revealed that a lower socioeconomic status significantly increased their risk, highlighting the critical link between socioeconomic disadvantage and heightened susceptibility to a potentially fatal yet preventable infection. Consequently, public health efforts should be made to understand the challenges facing those at greatest risk of melioidosis and to develop effective ways to disseminate information aiming to decrease the risk of both infection and subsequent disease. A study in Thailand established that simply imparting information does not necessarily transform daily practices to follow recommended guidelines [13]. The reluctance to change was attributed to deeply ingrained traditional behaviours, time constraints, and economic limitations—especially for not adopting protective footwear. Melioidosis awareness campaigns are conducted in the NT prior to each wet season through the local Centre for Disease Control (NTCDC). They distribute factsheets, posters, and radio announcements in English and five First Nation languages to people at risk of acquiring melioidosis [14]. One year, this included supplying enclosed shoes to particularly vulnerable populations [15]. Nevertheless, whether these programs have been successful in reaching certain populations who are at high risk of melioidosis is not known.

Current preventative measures recommended for at-risk people include wearing appropriate footwear and protective clothing, staying indoors during severe storms, ensuring wound coverage, and managing underlying chronic conditions like diabetes [16]. We designed our study with the objective of assessing participant awareness of established risk factors and protective measures for melioidosis. We wanted to know if people knew of both the exposure risks, such as performing outdoor activities without shoes and gardening during the wet season, and the clinical risks, such as diabetes, excessive alcohol use, and smoking. We carried this out by developing a questionnaire that assessed how aware individuals were of these factors. Diabetes has long been established as being one of the highest risk factors for melioidosis infection [17] The knowledge of this factor was assessed by asking if “Drinking sugary soft drinks” increased the risk of getting melioidosis. We additionally included less common risks, such as using “Needles for injection” and “Drinking tap water”. Although a significant risk in countries without chlorinated drinking tap water, drinking tap water in Australia has negligible risk unless supplied from a bore delivering unchlorinated water. Melioidosis from contaminated potable water has been documented in remote Aboriginal communities in northern Australia [18,19,20]. Similarly, though using needles for injections could be a hypothetical transmission mode, as is laboratory-acquired infection, the risks are considered extremely low [21].

As of the writing of this article, there has been no assessment of melioidosis knowledge within the NT population. Therefore, our study aimed to address this by objectively evaluating the level of melioidosis understanding among three specific groups in the NT, assessing their utilisation of preventative measures, and identifying more effective communication methods through participant feedback.

## 2. Materials and Methods

### 2.1. Ethics Considerations

The Human Research Ethics Committee of NT Health and Menzies School of Health Research granted approval for the study (HREC 21-4958). This included approval of a questionnaire designed to ensure data comprehension, extractability, and cultural safety. Participant consent was acquired electronically from those who completed the online questionnaire. Consent was obtained verbally and recorded for those who utilised the paper version. Data collection was from August 2021 to June 2023.

### 2.2. Participant Inclusion and Exclusion

For participant selection, three distinct groups were identified, as described below.

Group 1 (Hi-Ed): This group comprised individuals hypothesised to possess substantial knowledge about melioidosis prevention. Medical students from Flinders University in the NT and staff members of Menzies School of Health Research (Menzies) aged over 18 years were included. Their participation was contingent on providing consent to respond to the questionnaire.

Group 2 (Rangers/AHWs): This group comprised individuals recognised as trusted community professionals within First Nation populations, including four NT Aboriginal Ranger groups and Aboriginal Health Workers from one clinic in Katherine. Eligibility criteria were being aged over 18 years and providing consent to participate in completing the questionnaire.

Group 3 (Patients): This group comprised patients with a history of previously or recently diagnosed culture-confirmed melioidosis. We used this group to assess the awareness of knowledge dissemination within the hospital system. Eligibility criteria for inclusion encompassed patients aged over 18 years who were either attending Infectious Disease clinics or admitted to wards at Royal Darwin Hospital. Their consent was needed to participate in completing the questionnaire.

#### Exclusion Criteria

We excluded patients deemed unfit for participation by their treating medical teams and participants with language barriers, as translators were not used for this study.

### 2.3. Data Collection Procedure Using Questionnaire

The ethics-approved questionnaire (Appendix A) was developed through collaborative input from Infectious Disease Physicians, First Nation researchers, and Academics to ensure simple data handling and cultural safety. Several questions were formulated in line with a quantitative questionnaire designed by Chansrichavala et al. (2015) that studied the knowledge awareness of melioidosis in Thailand’s populations [13]. Surveys were administered online through the Qualtrics XMTM software (July 2020), taking approximately five minutes to complete. A printed questionnaire version was used when electronic devices were unavailable. In such cases, the researchers manually entered the data into the database following the completion of the questionnaire by the participant.

### 2.4. Survey Measures

The initial survey components included gender, age, duration of NT residency, postal code/region, study group classification, and the highest level of education attained. Following this, a question asked participants whether they had heard of melioidosis, but participants continued to answer the remainder of the questionnaire even if their response was negative. Two sets of questions employing a 3-point scale (yes/no/do not know) were presented. The first set assessed participants’ familiarity with various melioidosis risk factors (exposure risks and clinical risk factors), while the second evaluated their awareness of protective factors. For analysis, responses of “do not know” were consolidated under “no”.

Clinical risk factors evaluated included diabetes (assessed by asking about increased melioidosis risk due to consuming sugary soft drinks), alcohol consumption, and cigarette smoking. Exposure risks involved activities like gardening, hunting, and playing football during the wet season, as well as walking barefoot. Additionally, potential risks like drinking tap water and sharing needles were examined for comparative risk perception. These questions helped calculate the Risk Knowledge Index (RKI) score (see Section 2.4.1 Indices for assessing melioidosis knowledge).

Protective factors evaluated included overall health management, asking if consuming fruits and vegetables and reducing alcohol intake could prevent melioidosis. For exposure risk protection, factors like wearing boots and gloves outdoors, shoes when walking, covering wounds, staying indoors during storms, and wearing a mask while using a high-pressure hose were considered. These questions contributed to calculating the Protective Knowledge Index (PKI) score (see Section 2.4.1 Indices for assessing melioidosis knowledge).

Next, participants answered three multiple-choice questions. The first asked how they learned about melioidosis, allowing for multiple responses. The second inquired about sources for additional melioidosis information, also permitting multiple answers. These responses contributed to creating an Information Source Index (ISI) to investigate the link between the number of information sources used and melioidosis knowledge (see Section 2.4.1 Indices for assessing melioidosis, ISI (Information Source Index)). The final question in the questionnaire asked participants to select sources that could best help communicate the message of melioidosis prevention in their communities.

#### 2.4.1. Indices for Assessing Melioidosis Knowledge

Risk Knowledge Index (RKI): Defined as the additive sum of all risk factors for melioidosis (alcohol consumption; walking barefoot; playing football on wet grounds, gardening, and hunting in wet conditions; sharing needles; smoking cigarettes; drinking sugary soft drinks; and tap water), yielding scores from 0 to 9. Higher scores reflect greater knowledge of melioidosis risk factors.

Protective Knowledge Index (PKI): Defined as the additive sum of all protective factors, equally weighted (drinking less alcohol, wearing boots and gloves, eating fruit and vegetables, wearing masks when using a high-pressure hose, wearing shoes when outside, staying indoors during storms, covering wounds) to create values ranging from 0 to 7, where higher values represent greater levels of melioidosis protective factor knowledge.

Information knowledge index (ISI): Defined as the total of all sources used to learn about melioidosis, each weighted equally (including family/friend, radio, clinic/hospital, healthcare professionals, TV, newspaper, and posters), scoring from 0 to 7. Higher scores indicate the use of more information sources, with an extra point for any “other” unlisted sources. An additional point was given for any sources mentioned as “other” which were not provided as options.

Melioidosis Knowledge Index (MKI): Defined as the additive sum of knowledge about all risk factors (maximum 9), protective factors (maximum 7), and whether respondents had heard of melioidosis at all, with the question “Have you ever heard of melioidosis?” given a weight of 4. Higher values represent greater levels of overall melioidosis knowledge, with a maximum score of 20.

### 2.5. Statistical Analysis

Statistical analysis was conducted using R software (Version 4.2.2). Non-parametric tests analysed continuous variables due to non-normal data distribution. Linear regression models checked normality in residuals, not raw data. Categorical data were summarised in frequency tables and continuous data via the median and IQR. Fisher’s exact, Mann–Whitney U, and Kruskal–Wallis tests examined relationships and group differences. Spearman correlation assessed correlations between RKI, MKI, and PKI scores and various numeric variables like ISI, age, and years in NT. Univariate and multiple linear regressions identified MKI score predictors, incorporating the square root of ISI for better fitting. Multiple regression included significant univariate variables, checking residuals for patterns and outliers. Significance was set at *p* < 0.05.

## 3. Results

An interactive, detailed summary of our analysis can be accessed through this link: https://cnpdata.shinyapps.io/meli_project/ (accessed on 25 March 2024).

### 3.1. Baseline Characteristics (Table 1)

The distribution of gender did not exhibit significant differences across groups (P_Fisher exact_ = 0.118). Females constituted 59.6% of participants, with proportions of 64.8%, 48.0%, and 57.1% in Groups 1 (Hi-Ed), 2 (Rangers/AHWs), and 3 (Patients), respectively. Median age displayed significant variation across groups (P_Kruskal–Wallis_ < 0.001). Group 1 had a median age of 28.00 years (IQR: 22.00–36.25), Group 2 (Rangers/AHWs) had a median age of 34.50 years (IQR: 24.00–40.00), and Group 3 (Patients) had the highest median age of 50.00 years (IQR: 40.25–56.25). Educational attainment differed significantly (P_Fisher exact_ < 0.001). Group 1 (Hi-Ed) had the highest proportion with a bachelor’s degree at 52.0%, while Group 2 (Rangers/AHWs) and Group 3 (Patients) showed proportions of 6.1% and 3.7%, respectively. Group 1 (Hi-Ed) also had the highest postgraduate proportion at 34.4%.

### 3.2. Questionnaire Responses (Table 2)

Regarding the question “Have you ever heard of melioidosis?”, significant differences were observed among the groups (P_Fisher exact_ = 0.001). Group 1 (Hi-Ed) displayed the highest percentage, with 93.6% responding affirmatively, compared to 72.0% in Group 2 (Rangers/AHWs) and 81.5% in Group 3 (Patients).

**Table 2 tropicalmed-09-00071-t002:** Response to survey analysis.

Variable	Level	Overall *n* = 203	Group 1 (Hi-Ed) *n* = 125	Group 2 (Rangers/AHWs) *n* = 50	Group 3 (Patients) *n* = 28	*p*-Value
**Have you ever heard of melioidosis? (%)**	No	27 (13.4)	8 (6.4)	14 (28.0)	5 (18.5)	**0.001 *§**
	Yes	175 (86.6)	117 (93.6)	36 (72.0)	22 (81.5)	
**Do you think these behaviours increase your chance of getting melioidosis?**						
**1. Gardening in wet season (%)**	Don’t Know	26 (12.8)	8 (6.4)	15 (30.0)	3 (10.7)	**<0.001 *§**
	No	6 (3.0)	1 (0.8)	5 (10.0)	0 (0.0)	
	Yes	171 (84.2)	116 (92.8)	30 (60.0)	25 (89.3)	
**2. Hunting out bush during wet season (%)**	Don’t Know	35 (17.3)	13 (10.5)	17 (34.0)	5 (17.9)	**0.001 *§**
	No	12 (5.9)	4 (3.2)	5 (10.0)	3 (10.7)	
	Yes	155 (76.7)	107 (86.3)	28 (56.0)	20 (71.4)	
**3. Playing football on community oval in wet season (%)**	Don’t Know	33 (16.3)	13 (10.4)	13 (26.0)	7 (25.0)	**0.029 *§**
	No	15 (7.4)	7 (5.6)	5 (10.0)	3 (10.7)	
	Yes	155 (76.4)	105 (84.0)	32 (64.0)	18 (64.3)	
**4. Drinking sugary soft drinks (%)**	Don’t Know	48 (23.8)	24 (19.4)	17 (34.0)	7 (25.0)	**0.042 *§**
	No	94 (46.5)	66 (53.2)	20 (40.0)	8 (28.6)	
	Yes	60 (29.7)	34 (27.4)	13 (26.0)	13 (46.4)	
**5. Drinking tap water (%)**	Don’t Know	53 (26.1)	24 (19.2)	21 (42.0)	8 (28.6)	**0.011 *§**
	No	122 (60.1)	86 (68.8)	20 (40.0)	16 (57.1)	
	Yes	28 (13.8)	15 (12.0)	9 (18.0)	4 (14.3)	
**6. Smoking cigarettes (%)**	Don’t Know	55 (27.2)	32 (25.8)	19 (38.0)	4 (14.3)	0.197 §
	No	82 (40.6)	50 (40.3)	17 (34.0)	15 (53.6)	
	Yes	65 (32.2)	42 (33.9)	14 (28.0)	9 (32.1)	
**7. Drinking alcohol (%)**	Don’t Know	56 (27.7)	32 (25.6)	19 (38.8)	5 (17.9)	0.185 §
	No	66 (32.7)	45 (36.0)	13 (26.5)	8 (28.6)	
	Yes	80 (39.6)	48 (38.4)	17 (34.7)	15 (53.6)	
**8. Walking without shoes (%)**	Don’t Know	27 (13.3)	10 (8.0)	15 (30.0	2 (7.1)	**0.001 *§**
	No	9 (4.4)	4 (3.2)	2 (4.0)	3 (10.7)	
	Yes	167 (82.3)	111 (88.8)	33 (66.0)	23 (82.1)	
**9. Sharing needles for injections (%)**	Don’t Know	60 (29.7)	29 (23.4)	19 (38.0)	12 (42.9)	**0.03 *§**
	No	72 (35.6)	54 (43.5)	11 (22.0)	7 (25.0)	
	Yes	70 (34.7)	41 (33.1)	20 (40.0)	9 (32.1)	
**Do you think these actions protect you from getting melioidosis?**						
**1. Wearing boots and rubber gloves when working outdoors (%)**	Don’t Know	21 (10.3)	6 (4.8)	14 (28.0)	1 (3.6)	**<0.001 *§**
	No	4 (2.0)	1 (0.8)	3 (6.0)	0 (0.0)	
	Yes	178 (87.7)	118 (94.4)	33 (66.0)	27 (96.4)	
**2. Wearing shoes when walking outside (%)**	Don’t Know	20 (9.9)	7 (5.6)	13 (26.0)	0 (0.0)	**<0.001*§**
	No	3 (1.5)	1 (0.8)	2 (4.0)	0 (0.0)	
	Yes	180 (88.7)	117 (93.6)	35 (70.0)	28 (100.0)	
**3. Covering wounds with band-aids (%)**	Don’t Know	23 (11.4)	8 (6.5)	12 (24.0)	3 (10.7)	**0.021 *§**
	No	3 (1.5)	2 (1.6)	1 (2.0)	0 (0.0)	
	Yes	176 (87.1)	114 (91.9)	37 (74.0)	25 (89.3)	
**4. Staying indoors during storms (%)**	Don’t Know	35 (17.3)	21 (16.9)	13 (26.0)	1 (3.6)	0.082 §
	No	26 (12.9)	17 (13.7)	7 (14.0)	2 (7.1)	
	Yes	141 (69.8)	86 (69.4)	30 (60.0)	25 (89.3)	
**5. Drinking less alcohol (%)**	Don’t Know	55 (27.4)	34 (27.6)	16 (32.0)	5 (17.9)	0.708 §
	No	45 (22.4)	26 (21.1)	11 (22.0)	8 (28.6)	
	Yes	101 (50.2)	63 (51.2)	23 (46.0)	15 (53.6)	
**6. Eating lots of fruit and vegetables (%)**	Don’t Know	51 (25.4)	31 (25.2)	17 (34.0)	3 (10.7)	**0.04 *§**
	No	39 (19.4)	29 (23.6)	7 (14.0)	3 (10.7)	
	Yes	111 (55.2)	63 (51.2)	26 (52.0)	22 (78.6)	
**7. Wearing a mask when using a high-pressure hose**	Don’t Know	52 (25.9)	27 (22.0)	19 (38.0)	6 (21.4)	0.086 §
	No	15 (7.5)	11 (8.9)	4 (8.0)	0 (0.0)	
	Yes	134 (66.7)	85 (69.1)	27 (54.0)	22 (78.6)	
**Resources they used to hear about melioidosis (%)**	Less than 2	150 (73.9)	86 (68.8)	40 (80.0)	24 (85.7)	0.097 §
	2 or more	53 (26.1)	39 (31.2)	10 (20.0)	4 (14.3)	
**Do you know the NT government website that has a melioidosis also known as melioidosis factsheet (%)**	No	136 (67.0)	74 (59.2)	39 (78.0)	23 (82.1)	**0.011 *§**
	Yes	67 (33.0)	51 (40.8)	11 (22.0)	5 (17.9)	

§ Fisher exact test; * statistically significant results (*p* < 0.05).

There were significant differences among the groups (P_Fisher exact_ < 0.050 for all) for behaviours perceived to increase the risk of contracting melioidosis. Group 1 (Hi-Ed) consistently exhibited the highest proportions of participants correctly identifying risk-increasing behaviours. However, certain items such as “Drinking sugary soft drinks” (29.7%), which was a proxy for the risk of diabetes, “Drinking alcohol” (36%), and “Smoking cigarettes” (32.2%) highlighted lower proportions of participants responding correctly in Group 1 (Hi-Ed), indicating knowledge gaps even within this group (Table 2). In Group 2 (Rangers/AHWs), the highest score was seen in identifying the risk of “Walking without shoes”, where 66% answered correctly, while the lowest was “Drinking tap water” (18%). 

For protective measures, significant differences were found between the groups for various questions (“Wearing boots when working outdoors” (P_Fisher exact_ < 0.001), “Wearing shoes when walking outside” (P_Fisher exact_ < 0.001), “Covering wounds with band-aids” (P_Fisher exact_ = 0.021), and “Eating fruits and vegetables” (P_Fisher exact_ = 0.04)). We found that Group 3 (Patients) consistently scored slightly higher than Group 1 (Hi-Ed) in most questions (Table 2). Group 2 (Rangers/AHWs) scored lower than the other groups across all questions.

### 3.3. Melioidosis Knowledge Score Indices (Table 2 and Table 3)

There was no significant difference in RKI (Risk Knowledge Index) and MKI (Melioidosis Knowledge Index) scores across the groups. Group 3 (Patients) scored slightly higher for the MKI with 15 (12, 16.5), compared to Group 2 (Rangers/AHWs) with 13 (4.75, 16) and Group 1 (Hi-Ed) with 14 (12, 17). However, a statistically significant difference was found in the PKI (Protective Knowledge Index) (P_Kruskal–Wallis_ = 0.036), with Group 3 (Patients) having the highest score with a median (IQR) of 6 (5, 7) compared to a median (IQR) of 5 (4, 7) for Group 1 (Hi-Ed) and 5 (2, 6.76) for Group 2 (Rangers/AHWs). Education did not show a significant association with any scores. Interestingly, the number of resources used displayed the strongest association with knowledge scores. Those using two or more resources consistently scored higher on the RKI (P_Kruskal–Wallis_ = 0.008), PKI (P_Kruskal–Wallis_ = 0.013), and MKI (P_Kruskal–Wallis_ = 0.006) (Table 2) (Figure 1A). This was further supported by correlation analysis, where the number of resources used significantly correlated with the RKI (rho = 0.13, *p* ≤ 0.001), MKI (rho = 0.35, *p* < 0.0001), and PKI (rho = 0.34, *p* < 0.0001). No significant correlations were found for age or years lived in NT with any indices (Table 3) (Figure 1B–D).

**Table 3 tropicalmed-09-00071-t003:** Correlation between RKI, MKI, and PKI scores and different variables.

	RKI		MKI		PKI	
	Rho	*p*-Value	Rho	*p*-Value	Rho	*p*-Value
**ISI**	0.31	**0.001 µ**	0.39	**<0.001 µ**	0.34	**<0.001 µ**
**Age**	−0.039	0.57 µ	0.01	0.88 µ	0.14	**0.03 µ**
**Years lived in NT**	0.06	0.37 µ	0.05	0.43 µ	0.2	0.089 µ

µ Spearman correlation; RKI: Risk Knowledge Index; PKI: Protective Knowledge Index; MKI: Melioidosis Knowledge Index.

### 3.4. Regression Analysis (Table 4)

Lastly, a linear regression model revealed that education (β = 1.66; 95% CI: 0.28, 3.04; *p* = 0.049), ISI score (β = 4.12; 95% CI: 3.13, 5.12; *p* = 0.0019), and Group 2 (Rangers/AHWs) (β = −2.84; 95% CI: −4.45, −1.2; *p* = 0.002) significantly predicted MKI score in univariate analyses. However, in a multiple regression model accounting for factors that were statistically significant in the univariate model, only the ISI score emerged as a predictive variable (β = 3.79; 95% CI: 2.77, 4.80; *p* ≤ 0.0001), consistent with our initial results (Table 4).

**Table 4 tropicalmed-09-00071-t004:** Linear regression for factors predicting MKI.

Model 1 (Univariate)		
**Variable**	**Coefficient (95% CI)**	** *p* ** **-Value**
**Gender (Male)**	−1.01 (2.33, 4.11)	0.16
**Age**	0.016 (−0.04, 0.071)	0.57
**Years lived in NT**	0.0016 (−0.042, 0.045)	0.054
**Education (Bachelors) †**	1.66 (0.28, 3.04)	**0.018 ***
**ISI score §**	4.12 (3.13, 5.12)	**<0.001 ***
**Group (2) #**	−2.84 (−4.45, −1.2)	**0.001 ***
**Group (3) #**	0.02 (−2.01, 2.06)	0.98
**Model 2 (Multivariable) µ**		
**Education (Bachelors)**	0.08 (−1.71, 1.87)	0.86
**ISI score §**	3.79 (2.77, 4.80)	**<0.001 ***
**Group (2) #**	−1.44 (−3.38, 0.50)	0.25
**Group (3) #**	0.65 (−1.59, 2.91)	0.34

* Statistically significant results (*p* < 0.05); † reference group is non-bachelor; § we used the square root of ISI to improve model fitting; # reference group is group 1; µ accounted for education, ISI score, and group.

### 3.5. Results for Communication Modalities (Appendix A)


**How have you heard about melioidosis?**


The sources through which individuals have acquired information about melioidosis reveal intriguing patterns across the three groups. For Group 1 (Hi-Ed), the primary information sources were “Family members or friends” (38%), followed by “Clinic or hospital visits” (29%), and “Posters” (19%). For Group 2 (Rangers/AHWs), “I do not know what this is” (32%) was the most common answer, followed by “From a family member or friend” (28%), and “From my doctor or healthcare worker” (16%). Group 3 (Patients) demonstrated another shift in preferences with “Clinic or hospital visits” (29%) emerging as the primary source of information, aligning with them hearing about melioidosis when they first presented sick with it, followed by “Healthcare providers” (18%) and “Posters” (14%).


**Where did you go to find more information about melioidosis?**


The responses to this question elucidate information-seeking behaviours within the surveyed groups. In Group 1 (Hi-Ed), “Internet searches” (67%) constituted the predominant method, followed by “Doctors or healthcare workers” (12%) and “Pamphlets in clinics or hospitals” (9%). In Group 2 (Rangers/AHWs), “Doctor or healthcare worker” consultations consisted of 34%, followed by “Internet searches” (22%) and “Reading a pamphlet in the clinic or hospital” (18%). Notably, Group 3 (Patients) displayed a distinct preference for relying primarily on “Doctors and healthcare workers” (43%) for information, followed by the “Internet” (21%) and “Family and friends” (14%).


**What do you think is the best way to help people in your community avoid getting melioidosis?**


The responses to this question exhibit variations in preferences for melioidosis prevention strategies among the three distinct groups. Notably, within Group 1 (Hi-Ed), the top three preferred methods include “Doctors and healthcare workers giving advice” (70%), “At schools so that my kids hear about it” (68%), and “TV advertising” (62%). Similar trends emerge for Group 2 (Rangers/AHWs) with “Doctors and healthcare workers giving advice” (70%), “At schools so that my kids hear about it” (62%), and “Radio” (60%) ranking as the top three choices. Conversely, Group 3 (Patients) demonstrated a distinct preference, with the highest being “Doctors and healthcare workers giving advice” (75%), followed by “TV advertising” (62%) and “At schools so that my kids hear about it” (46%).

## 4. Discussion

This study is the first to investigate melioidosis awareness in the NT, a region with high melioidosis incidence. It focused on NT medical students and Menzies School of Health Research staff (Group 1), Aboriginal Rangers and Healthcare Workers (Group 2), and melioidosis-diagnosed hospital patients (Group 3). We observed awareness gaps in all groups regarding risk and protective factors. Group 1 was more aware of melioidosis than Groups 2 and 3, with better knowledge of risk factors but limited awareness of key risks like diabetes, alcohol, and smoking. Group 3 showed a higher awareness of protective factors, likely due to hospital education.

Analysis of the questions regarding communication modalities used to obtain more information about melioidosis revealed valuable results. We found that the number of sources used to hear about melioidosis significantly increased the overall melioidosis knowledge in participants, whilst education, group, and number of years lived in the NT did not. In addition, the modality deemed most useful for effective dissemination of melioidosis prevention knowledge in communities was through “Doctors and healthcare workers giving advice” followed closely by “At school so that my kids can hear about it”.

A study with a similar objective conducted by Smith et al. (2022) surveyed hospital patients with risk factors for melioidosis in Far North Queensland (FNQ) [11]. Both this study and ours in the NT have revealed gaps in awareness about melioidosis infection within high-risk populations. However, the overall percentage of participants who reported having heard about melioidosis in our study was higher (175 of 202 or 86.6%) compared to the population studied in FNQ (29 of 361 or 19%) [11]. These differences are likely due to the lower incidence of melioidosis in FNQ, targeted respondent selection, and more effective public health campaigns in the NT [15].

Our study highlights that education alone may not guarantee an in-depth understanding of melioidosis and that targeted interventions should be considered to further educate those living in melioidosis-endemic regions. A previous study in Indonesia also revealed that knowledge about melioidosis is limited among their healthcare workers [22]. Although that study differs from ours in that they tested knowledge about diagnostics and the treatment of melioidosis, it demonstrates the need for more knowledge provision within groups responsible for disseminating health information to future at-risk populations [22].

Group 3 (Patients) had the most significant protective factor knowledge (PKI) level compared to both groups. In fact, their level of risk knowledge (RKI) and overall melioidosis knowledge (MKI) were on par with those of Group 1 (Hi-Ed), which was considered to be our high-awareness group. This is a positive finding and validates the education provided through the hospital system.

One of our study’s key findings is that the number of resources used to learn about melioidosis increased participant knowledge levels pertaining to this infection. This further highlights the need for a diversified range of information sources being available. Public health campaigns could leverage various media channels, health platforms, and community engagement strategies to empower individuals to make informed choices. By providing easily accessible and trustworthy information, these initiatives can bridge knowledge gaps and promote healthier behaviours. A study carried out by D’Coasta et al. (2022) to evaluate the impact of syphilis public health campaign messaging in remote parts of Australia found that, at least for the younger demographics (age 15 to 29 years), multi-media modalities deemed be the most effective included television, Facebook, and websites on the Internet. These were reported as the most popular sources of campaign exposure routes [23].

Public health messaging aimed at preventing melioidosis among individuals with risk factors involves the recommendation to wear shoes when outdoors to reduce the risk of inoculating injuries and to seek shelter indoors during storms to minimise the chances of inhaling *B. pseudomallei*. These messages have been translated into five First Nation languages, including Eastside Kriol and Warlpiri, and are broadcasted on local radio stations during each wet season [24]. Nevertheless, our study revealed that radio may not be the most effective way to reach this population. When answering the question “What do you think is the best way to help people in your community avoid getting melioidosis?”, the most recommended method by all groups was “Doctors and healthcare workers giving advice”, followed by “At schools so that my kids hear about it”, with “Radio advertising” coming in fifth place. Aboriginal people have a long history of traditional storytelling as a way of sharing information. This finding reinforces the need to provide information in a way that meets community needs. Furthermore, a previous study carried out in Thailand looking at the barriers to following preventative measures by people at risk of getting melioidosis found that providing information alone is unlikely to make a change, due to the inherent tradition of some behaviours such as barefoot rice planting [10]. Future similar research exploring barriers in the NT context will need to be conducted to help us understand these obstacles. These could then be fed into campaigns attempting to provide feasible protective actions and education.

There are limitations recognised by the authors of this study that warrant consideration. Firstly, while stratified into three distinct groups, the participant selection process lacked randomization and was a convenience sample, potentially introducing sampling bias. Notably, medical students and staff members of Menzies School of Health Research included in Group 1 may possess higher baseline knowledge of melioidosis prevention than the broader educated population, thereby affecting the generalizability of findings. Secondly, reliance on self-reported data collected through questionnaires raises concerns about social desirability bias, wherein participants may provide responses aligned with societal expectations rather than their actual knowledge. Thirdly, the study predominantly focused on demographic and information source variables in its correlation and regression analyses, overlooking potential confounders such as socioeconomic factors, which could provide a more comprehensive understanding of melioidosis knowledge. Fourthly, the absence of information regarding response rates poses a significant challenge in evaluating the adequacy of the sample size and raises concerns about the potential introduction of selection bias. Also, while the questionnaire aimed to assess participants’ perceptions regarding behaviours that may influence the risk of melioidosis, including smoking, alcohol consumption, and sugary drink intake, it is acknowledged that the direct causal relationship between these behaviours and melioidosis is not firmly established in the scientific literature. Instead, we used these as proxies for participants’ knowledge about the studied risk factors. For example, sugary drink intake was used as a proxy for knowledge of diabetes increasing the risk of melioidosis infection. Future surveys should incorporate more targeted questions to enhance specificity and accuracy in assessing participants’ knowledge and perceptions. Another limitation of the survey’s design is the absence of negatively framed or detractive questions, which could have provided a more nuanced understanding of participants’ knowledge and awareness levels. Negative questions, such as identifying behaviours that do not increase the risk of melioidosis or actions that are not protective measures, could have revealed misconceptions or gaps in understanding among respondents. Lastly, given that no translators were used in this study, the potential influence of language factors on questionnaire interpretation and responses must be acknowledged, a facet insufficiently explored in this study. Equally, it is likely that people without formal schooling or schooling in languages other than English have been entirely excluded, thus introducing further bias and generalisability to the NT population. A study of Yolŋu patients receiving dialysis in Darwin showed that significant misunderstandings of health communication occur between Aboriginal patients and their healthcare providers, even when Aboriginal health workers were involved [25,26]. Conversely, the provision of Aboriginal Interpreters was found to have transformed the experience and understanding of medical information provided to them when this service had been embedded within the renal unit at Royal Darwin Hospital, demonstrating the unparalleled value of their contribution [27].

## 5. Conclusions

Our findings have several implications for public health interventions addressing melioidosis. First, there is a need for targeted awareness campaigns, especially among Australian First Nation populations at higher risk due to occupational exposure and higher burden of co-morbid disease. Secondly, gaps in knowledge within high-awareness groups need to be addressed as individuals within this group will become responsible for the eventual distribution of knowledge to the public. These campaigns should address specific knowledge gaps and utilise multiple communication channels to reach diverse audiences effectively. Thirdly, further studies are required to look at the barriers to adopting measures to prevent infection with *B. pseudomallei* in northern Australia to design the most effective public health interventions for addressing melioidosis.

## Figures and Tables

**Figure 1 tropicalmed-09-00071-f001:**
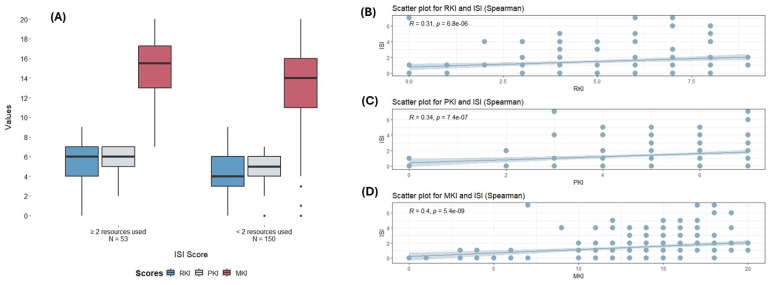
(**A**) Comparison of different scores (RKI, PKI, MKI) according to the number of melioidosis resources used (≥2 recourses or <2 resources). Dots represent outlier points. (**B**–**D**): Correlation between the number of resources used and the different knowledge scores. RKI: Risk Knowledge Index; PKI: Protective Knowledge Index; MKI: Melioidosis Knowledge Index; ISI: Information Source Index.

**Table 1 tropicalmed-09-00071-t001:** Baseline summary of included individuals (percentage).

Variable	Level	Overall *n* = 203	Group 1 (Hi-Ed) *n* = 125	Group 2 (Rangers/AHWs) *n* = 50	Group 3 (Patients) *n* = 28	*p*-Value
**Gender (%)**	Female	121 (59.6)	81 (64.8)	24 (48.0)	16 (57.1)	0.118 §
	Male	82 (40.4)	44 (35.2)	26 (52.0)	12 (42.9)	
**Age (median [IQR])**		30.00 [24.00, 42.00]	28.00 [22.00, 36.25]	34.50 [24.00, 40.00]	50.00 [40.25, 56.25]	**<0.001 *†**
**Age (%)**	≤20	23 (11.4)	18 (14.5)	5 (10.0)	0 (0.0)	**<0.001 *§**
	21–30	78 (38.6)	61 (49.2)	14 (28.0)	3 (10.7)	
	31–40	43 (21.3)	21 (16.9)	18 (36.0)	4 (14.3)	
	41–50	34 (16.8)	17 (13.7)	8 (16.0)	9 (32.1)	
	≥51	24 (11.9)	7 (5.6)	5 (10.0)	12 (42.9)	
**Years live in NT (median [IQR))**		20.00 [7.00, 28.50]	12.00 [5.00, 21.00]	29.00 [22.00, 40.00]	48.00 [22.50, 51.25]	**<0.001 *†**
**Education (%)**	Bachelor’s degree	69 (34.3)	65 (52.0)	3 (6.1)	1 (3.7)	**<0.001 *§**
	Grade 12 or apprenticeship	56 (27.9)	16 (12.8)	33 (67.3)	7 (25.9)	
	Grade 9 or less	28 (13.9)	1 (0.8)	10 (20.4)	17 (63.0)	
	Postgraduate	48 (23.9)	43 (34.4)	3 (6.1)	2 (7.4)	

§ Fisher’s exact test; † Kruskal–Wallis test; * statistically significant results (*p* < 0.05).

## Data Availability

The datasets used and/or analysed during the current study are available as MS excel files from the first author upon a reasonable request.

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
