# Peer review of "Melioidosis Knowledge Awareness in Three Distinct Groups in the Tropical Northern Territory of Australia"

_tropicalmed, 2024, doi:10.3390/tropicalmed9040071_

Round 1

Reviewer 1 Report

Comments and Suggestions for Authors

The study is relevant since it discusses education in melioidosis, a priority topic in the approach to a disease that has important implications for public health.

Objective and clear manuscript, that manages to show the need for attention to the topic. Appropriate methodology considering the limitations that were well presented in the discussion.

Small suggestions:

# 131 Exclusion criteria: remove items: age under 18 and non-consent. Exclusion criteria comprehend impediments to participation, despite the individual meeting the inclusion criteria.

#409 In the last two paragraphs there is a repetition of the conclusion.

Reviewer 2 Report

Comments and Suggestions for Authors

Thank you for the opportunity to review this manuscript.

This study utilised a survey to identify to assess melioidosis knowledge among 3 different groups in the Northern Territory of Australia. This work is important as identifying gaps in melioidosis awareness may help inform public health campaigns that aim to reduce the incidence of the disease. 

I have several minor comments.

How was the survey provided to the participants and how did you identify the participants? Was this by emailing the questionnaire?

I appreciate the need to develop the questionnaire in understandable terms, however I wonder if there is a risk that these could be misinterpreted, particularly in people with knowledge of the disease. Although smoking may lead to the development of chronic lung disease, not all people who smoke develop chronic lung disease and I am unaware that smoking itself is a risk factor for melioidosis. Similarly, drinking alcohol in non-hazardous amounts does not appear to increase the risk of melioidosis. Drinking sugary soft drinks has been associated with obesity and in many studies, an increased risk of type II diabetes, but it is unclear if this is related to increased caloric intake and thus weight gain. While I appreciate this is unable to be changed at this stage, addition of this limitation to the discussion may be helpful.

In the discussion, the authors mention that there was limited awareness of key risks like diabetes, alcohol, and smoking. Unfortunately, this relies on the assumption that drinking sugary soft drinks equates to diabetes in all people which it does not.

Line 337 compares a study from FNQ to the current study and hypothesises that the difference is melioidosis awareness is due to a lower incidence of FNQ. While this may be true, there are likely to be other reasons including the targeted respondents and more effective public health campaigns that occur in the NT.

There is limited data to support current public health messaging around public health messaging in melioidosis and therefore these may not be “crucial” as written in line 367.

Awareness and knowledge is gauged on the term melioidosis. I would be interested to know if people in the NT might be more familiar with other names for the disease, such as Nightcliff gardener’s disease or similar?

Some of the questions on the questionnaire include “tick all that apply”, however some don’t. Is there a risk that on some of the questions, people may have ticked one box only?

Comments on the Quality of English Language

There a couple of very minor typos

Line 47 “still still-increasing”

Line 111. “June 20233”

Line 162 “freducing”

Line 168 “Participants”

Reviewer 3 Report

Comments and Suggestions for Authors

This piece of work will be important to the community in Australia, that may also instruct other endemic regions in terms of their population awareness of melioidosis.

One issue I find with the survey is that the questions on sugary drinks and alcohol are ambiguous. It suggests that there is a direct link to melioidosis, without phrasing it more appropriately like these will cause you to have diabetes or liver problems that could make you get more severe melioidosis.  The interpretation of group 1 (highly ed) group shows more "wrong" answers but this could simply be interpreted as correct because the people do not think that drinking sugary drinks will give you melioidosis. In fact it doesn't. The results showing group 3 (patients) answering correctly could be attributed to bias and not necessarily understanding the link of these chronic diseases to melioidosis.

A second issue is that the survey questions are all in the affirmative. This means the "correct" answers are all yes. This could explain patients scoring higher. There were no detractive questions that were wrong, and so the patients would answer mostly in the affirmative. It may not necessarily mean the education they got in the hospitals now make them more aware. 

These caveats should be discussed.
